## [Reviewer comments · BMJ Open]

ARTICLE DETAILS

TITLE (PROVISIONAL)	Current practice of usual clinic blood pressure measurement in people with and without diabetes: a survey and prospective "mystery shopper" study in UK primary care
AUTHORS	Stevens, Sarah; McManus, Richard; Stevens, Richard

VERSION 1 – REVIEW

REVIEWER	George Stergiou Hypertension Center STRIDE-7, School of Medicine, University of Athens, Greece
REVIEW RETURNED	20-Nov-2017

GENERAL COMMENTS	This study investigated the office BP measurement methodology in primary care in UK by asking patients volunteers to participate in an online survey and provide retrospective information on office BP measurement during their last office visit and again to report prospectively about their next office visit. The main conclusion of the authors is that in most cases the routine BP measurement in UK primary care is in line with current UK guidelines. COMMENTS 1. There is a major selection bias in this study, which is evident by the high proportion of diabetics. In fact, this study can only discuss about (i) diabetics. (ii) who were involved in patient groups and (ii) were capable and also willing to participate.2. The response rate for initial survey was 34%. Eventually only 15% participated in the prospective survey. Thus, generalizing these results is tricky.3. An important message is that in this group of 'patients' 34% did not have their BP measured at their last appointment? Correct?4. The analysis of non-diabetics is based on a very small sample and the results are very uncertain. I believe this would be a much cleaner report if non-diabetics are discarded and data only on diabetics are presented.5. Other factors, such as time frequency of office visits, the presence of cardiovascular disease, the provision of home BP measurement may have also affected the office BP measurement methodology in some individuals.6. These data strictly apply to UK primary care and might considerably differ in other countries with different national healthcare systems.7. I am not sure whether the data presented in this study can lead to the second and the third point of the Abstract Conclusions, which refer to limitations of office BP measurement that have not been investigated in this study.
---

	8. Page 3: 'Hypertension trials and major epidemiological studies typically measure clinic BP using strict protocols on three or more times per visit'. In fact, in the vast majority of the cases 2-3 readings were taken and in several cases single measurements only. More than 3 readings per visit is rare.
--	---

REVIEWER	willem verberk Cardiovascular research institute Maastricht university, The Netherlands Microlife AG Switzerland
REVIEW RETURNED	13-Dec-2017

GENERAL COMMENTS	The authors have performed an interesting and useful study with the aim to determine the current practice of BP measurement during routine general practice appointments using a "mystery shopper model" General comments The conclusion: "current practice of routine BP measurement in UK primary care is concordant with guidelines in the majority of cases" sounds optimistic but I think the results of this study give less reason to be optimistic of how blood pressure measurement is performed and mentioning the limitation would give a more important clinical message. In addition, for reasons stated below is it not correct to mention that the "guidelines" are followed when mainly considering the number of blood pressure measurements The authors mainly focus on the number of blood pressure measurement taken. I think it is important to mention that this is just a part of the guidelines for blood pressure measurement. There are also other aspects that influence blood pressure values. For instance, who is measuring the blood pressure (nurse, GP) and e.g. for the sprint trial it is suggested that BP values were low because patients were left unattended and 5 minutes waiting time preceded the measurement. The authors mention in the introduction that it is important to understand how office blood pressure is measured for overall understanding. However, it is also important to understand if treatment was based on office blood pressure measurement. For instance, if blood pressure is high the physician can decide to prescribe ABPM or suggest to perform self BP measurement at home upon which the final treatment is based. Unfortunately, this does not become clear from the study as questions like: does your doctor verify your self-measured blood pressure values? did you get a 24-h blood pressure measurement? were missing. Minor comments: Regarding measurements according to the guidelines I presume that it is also Ok if the physician would take standard 3 measurements or take two measurements if the first one is below 140/90 mmHg. I wonder if there was a difference in average BP values between nurses or doctors taking BP measurements. For instance, was the 1st measurement obtained by the nurse likely to be lower than when taken by the doctor? It seems that the GP recommends Home BP more often than the nurse does. I think this is an important finding and worth mentioning in the discussion. Overall, I also think that patients are not often
--

	encouraged to measure their blood pressure at home, not even if they are hypertensive (approx.. 15%). Figure 2 shows that with 2 measurements (blue line) the second reading is much lower than the 1st which would indicate that a third measurement is needed. The green line shows that the 1st and 2nd reading are almost similar and thus there would not be the need for a third measurement. This is probably related to the low number of patients but still this seems in contradiction to what one would expect (following the guidelines). What is the reason for this? Discussion The authors mention “UK” guidelines. However, I also know the guideline from the BHS. Do GPs generally follow NICE or also BHS guidelines? If my understanding is right, there are some differences between these guidelines. The authors mention that the internet survey has the risk of underrepresenting certain groups such as e.g. the elderly (because of Internet). The survey requires a lot of reading and time to fill out which by itself also will lead to selection. As mentioned; considering guidelines for blood pressure measurement does not only entail the number of blood pressure measurements but also topics such as: was the right cuff size use, back support, no talking validated device measurement at heart level, no talking etc. etc. it is understandable that this could not all be asked but it is a limitation Masked hypertension is shortly mentioned but might be good to give an estimated prevalence of masked hypertension based on previous studies. It might not be the aim of the paper but 50% of the patients without diabetes are measured only one time. I would think this is quite concerning. I think the authors agree with this : “ GPs may be better advised to use multiple readings more widely”. There is a reason that “Hypertension trials and epidemiological studies use multiple clinic blood pressure measurements at each visit”
--	--

VERSION 1 – AUTHOR RESPONSE

Response to reviewers:

Reviewer 1

1. There is a major selection bias in this study, which is evident by the high proportion of diabetics. In fact, this study can only discuss about (i) diabetics. (ii) who were involved in patient groups and (ii) were capable and also willing to participate.

We agree with the reviewer that the population of patients that we recruited to collect data is selective. However, since the purpose of this study was to examine the behaviour of family doctors and nurses, any threat to the validity of our results would arise due to selection bias in the population of doctors. It is speculative to argue that patients who belong to user groups attend different doctors, or have their blood pressure measured differently, from patients who do not belong to user groups, although we cannot entirely rule this out (e.g. due to the inverse care law). It would also be surprising if family physicians treated those with diabetes very differently from others, although we agree with the reviewer that this is possible. It is for this reason that we present results stratified by diabetes status. Further, even if such biases at the patient-level also affected representation at the family physician level, previous research, regarding current practice in the use of BP self-monitoring, found similar results when using data from a convenience and representative sample of UK family doctors (see Fletcher et al. Br J Gen Pract 2016; 66 (652): e831-e837). We would therefore argue that the issue of

patient selection bias will result in limited biases in our study of doctor and nurse behaviour. We have, however, added further text to the discussion, addressing these issues.

"... Recruitment through patient involvement groups may have also resulted in over-representation of patients who are actively engaged with their healthcare, and due to phenomenon such as the inverse-care law,(24) may receive better quality (guideline adherent) care. However, since our aim was to study the behaviours of healthcare professionals, it is unclear how any biases at the patient-level will have translated into biases at the healthcare professional level. Furthermore, previous research regarding current practice of BP self-monitoring, showed similar results using both convenience and representative sampling of professionals. (25)"

2. The response rate for initial survey was 34%. Eventually only 15% participated in the prospective survey. Thus, generalizing these results is tricky.

As with our response to comment 1, we agree with the reviewer that there is selection bias in the patients reporting data, but would argue that this does not necessarily translate into biases in the healthcare professionals under study. We would also highlight that our study design, asking patients to report the behaviour of their health care practitioner, is novel. We selected this design over others in order to limit the effect of other potential biases, for example of self-reported data by surveying doctors directly. Due to limitations on word count, we are not able to discuss all of the study designs considered in detail. However, we have added the following text to the discussion section:

"Although our mechanism of data collection, asking patients to study the behaviour of their health care practitioners, has the limitations discussed above, we chose our "mystery shopper" approach over several other study designs. For a full discussion of the study designs considered see Stevens PhD thesis, 2017.(26) Briefly, studies based on alternative methodologies, such as practitioner self-report or direct observation, would have been subject to selection bias among practitioners, the Hawthorne effect, and reporting bias and we have avoided these biases through our novel design."

3. An important message is that in this group of 'patients' 34% did not have their BP measured at their last appointment? Correct?

The reviewer is correct that of those completing the prospective survey (n=334), 117 (35%) did not have their BP measured during their last appointment. We believe it is difficult to comment on the importance of this result within this study, which examines a snapshot of a single appointment and not the practice of BP monitoring over several appointments. In a single appointment, measurement of blood pressure may not be indicated depending on the presenting complaint (e.g. ingrown toenail) and we chose not to collect information about the reason for consultation as patients may have found it difficult to "pigeon-hole" the reason for consulting if several health problems are of concern. Furthermore, priorities may differ between the patient and clinician depending on perceived severity of the presenting problem(s), and the potential for opportunistic BP measurement is likely to be limited in a usual 10-minute appointment window.

4. The analysis of non-diabetics is based on a very small sample and the results are very uncertain. I believe this would be a much cleaner report if non-diabetics are discarded and data only on diabetics are presented.

We are reluctant to deviate from our original analysis plan, to the extent of discarding such a large amount of data (of those who had their BP measured, nearly 1 in 5 did not have diabetes). We agree with the reviewer that results for those without diabetes are uncertain, hence our post-hoc decision to stratify our results by diabetes status. This has allowed us to demonstrate, as far as possible, that this

limitation is unlikely to alter our main findings and we have no reason to believe practice would be different for those without diabetes.

5. Other factors, such as time frequency of office visits, the presence of cardiovascular disease, the provision of home BP measurement may have also affected the office BP measurement methodology in some individuals.

We agree with the reviewer and have addressed this issue for key factors such as diabetes/ hypertension/ treatment status in our analyses. Indeed, despite the overall observed patterns, there was considerable variability for individual patients. For example, in those with only one reading, systolic BP was as high as 213/100 mm Hg in one respondent when their BP was measured by a GP. Comparatively, one individual who had their BP measured three times by a nurse had an initial reading of 128/87 mm Hg. This suggests that for certain individuals, the BP reading alone does not dictate whether repeat readings are taken. This may be driven by the reason for the consultation (as discussed in our response to comment 3) or other patient factors. In this particular example, the respondent with only a single reading had no other cardiovascular risk factors or comorbidities, despite being older (71 years). The respondent who had their BP measured three times was younger (60 years) but had diabetes, rheumatoid arthritis, a history of heart attack and was taking both antihypertensive and statin medication. We have added further text to the discussion highlighting this point:

“Many factors, other than the initial BP value, can influence the decision to measure BP multiple times including previous measurement of BP in clinic or at home and the presence of cardiovascular disease or cardiovascular risk factors. Such factors may explain the considerable variability in BP measurement practices observed in some specific patient examples. Although we have addressed key factors such as diabetes, hypertension and treatment status, future research could explore behaviour in other subgroups. Furthermore, we did not ask respondents about the primary reason for their consultation which may have influenced BP measurement and this also requires further study.”

6. These data strictly apply to UK primary care and might considerably differ in other countries with different national healthcare systems.

We do not claim that our study results are generalizable beyond the UK, and agree that practice may be different in other healthcare systems. We have expanded some of the text regarding comparisons with existing literature to comment upon this:

“Our results support the idea that professionals treat single readings with caution, particularly those above the diagnostic threshold, which require further action (e.g. in the form of treatment change). Although numbers were smaller, results also suggest that this caution also extends to high BP sustained over two readings. Previous research suggests that recording of blood pressure may be influenced by specific BP related targets in the UK’s Quality and Outcomes Framework,(12) and hence routine practice in other healthcare systems, with different incentive schemes, may differ. Despite agreement between current practice and guidelines, GPs may be better advised to use multiple readings more widely,(28) to ensure comparability with BP monitoring studies and detection of masked hypertension which affects approximately 19% of adults.(29) ”

7. I am not sure whether the data presented in this study can lead to the second and the third point of the Abstract Conclusions, which refer to limitations of office BP measurement that have not been investigated in this study.

We agree with the reviewer and have edited the Abstract Conclusions as follows:

“Current practice of routine BP measurement in UK primary care is often concordant with guidelines for repeat measurement. Further studies are required to confirm findings in broader populations, to confirm when a third repeat reading is obtained routinely and to assess adherence to other aspects of BP measurement guidance.”

8. Page 3: ‘Hypertension trials and major epidemiological studies typically measure clinic BP using strict protocols on three or more times per visit’. In fact, in the vast majority of the cases 2-3 readings were taken and in several cases single measurements only. More than 3 readings per visit is rare

We thank the reviewer for highlighting this. We have amended the wording of this sentence and additionally referenced a 2014 review of hypertension trials which showed that 27% and 45% of trials measured BP on two and three or more occasions respectively.

“Hypertension trials and major epidemiological studies typically measure clinic BP using strict protocols on two to three times per visit in most cases.(3)”

Reviewer 2

The authors have performed an interesting and useful study with the aim to determine the current practice of BP measurement during routine general practice appointments using a “mystery shopper model”

General comments

1. The conclusion: “current practice of routine BP measurement in UK primary care is concordant with guidelines in the majority of cases” sounds optimistic but I think the results of this study give less reason to be optimistic of how blood pressure measurement is performed and mentioning the limitation would give a more important clinical message. In addition, for reasons stated below is it not correct to mention that the “guidelines” are followed when mainly considering the number of blood pressure measurements

We agree with the reviewer and have amended the Abstract conclusion as follows:

“Current practice of routine BP measurement in UK primary care is often concordant with guidelines for repeat measurement. Further studies are required to confirm findings in broader populations, to confirm when a third repeat reading is obtained routinely and to assess adherence to other aspects of BP measurement guidance.”

We have also amended the final paragraph of the discussion:

“The results of this study provide a preliminary insight into how BP is measured routinely and indicate that repeat BP measurements are taken in line with guidelines but not with strict study protocols.”

2. The authors mainly focus on the number of blood pressure measurement taken. I think it is important to mention that this is just a part of the guidelines for blood pressure measurement. There are also other aspects that influence blood pressure values. For instance, who is measuring the blood pressure (nurse, GP) and e.g. for the sprint trial it is suggested that BP values were low because patients were left unattended and 5 minutes waiting time preceded the measurement.

We agree that our focus on repeat BP measurement was not stated clearly enough. We have edited the introduction to better reflect our initial aims:

“We therefore sought to determine the current practice of BP measurement during routine appointments in UK primary care, focusing on when repeat clinic and home BP measurements are obtained.”

We have also edited some sections of the methods to make the focus on repeat clinic/ home monitoring more explicit:

Online survey: “Respondents anonymously reported basic demographic and health information, if and how many times their BP was measured at their last appointment and (recall permitting) their last BP reading (Supplement). Respondents were also asked about recommendations to monitor their BP at home.”

Statistical analysis: “The prospective study was powered to estimate the proportion of people having their BP measured once or multiple times, in line with guidelines at the 95% confidence level”

Finally, we have added text to the limitations section, which reflects on other aspects of BP measurement guidance:

“Guidance covers many factors affecting the accuracy of BP measurement, such as the use an appropriately sized cuff, but such factors are less easily assessed by patients and we chose to limit the focus of this study in order to maximise response rates. The type and accuracy of devices used in UK general practice has been studied previously,(27) but further direct observation of clinicians is warranted to determine if other aspects of BP measurement guidance is followed.”

3. The authors mention in the introduction that it is important to understand how office blood pressure is measured for overall understanding. However, it is also important to understand if treatment was based on office blood pressure measurement. For instance, if blood pressure is high the physician can decide to prescribe ABPM or suggest to perform self BP measurement at home upon which the final treatment is based. Unfortunately, this does not become clear from the study as questions like: does your doctor verify your self-measured blood pressure values? did you get a 24-h blood pressure measurement? were missing.

We agree that the basis of treatment is also an important research question, but we do not think it is one that can be well-captured by our patient-centric research methodology. As discussed above (response to reviewer 1) we selected our methodology to address our primary research question in a novel way.

Minor comments:

1. Regarding measurements according to the guidelines I presume that it is also Ok if the physician would take standard 3 measurements or take two measurements if the first one is below 140/90 mmHg.

We based our definition of “according to guidelines” based on the exact number of measurements recommended. Hence in the example given, such a patient would have been categorised as NOT having their BP measured according to guidelines. In those with/ without diabetes 14.3%/ 25.0% had there BP measured more than guidance recommends (as in the example). We have added a sentence to the methods section to make this clearer:

“Respondents who had their BP measured more or less than guidance recommends were deemed not to have had their BP measured according to guidance.”

2. I wonder if there was a difference in average BP values between nurses or doctors taking BP measurements. For instance, was the 1st measurement obtained by the nurse likely to be lower than when taken by the doctor?

We did not carry out this analysis as it is well established that BP is lower when measured by a nurse compared to a GP (see C Clark et al. Br J Gen Pract 2014; DOI: 10.3399/bjgp14X677851.) In our study, those with diabetes were more likely to have an appointment with a nurse (in most cases BP was measured by the same personnel conducting the appointment) and hence such an analysis is likely to be confounded by patient factors and unlikely to provide any meaningful insight into GP-nurse BP differences. However, we are happy to report for the reviewer that in the prospective study, in those reporting all BP readings, first systolic/ diastolic BP was 139/79 when measured by a GP and 132/77 when measured by a nurse, consistent with previous research.

3. It seems that the GP recommends Home BP more often than the nurse does. I think this is an important finding and worth mentioning in the discussion. Overall, I also think that patients are not often encouraged to measure their blood pressure at home, not even if they are hypertensive (approx.. 15%).

We thank the reviewer for this comment. We do not wish to interpret this finding too strongly because of the potential for confounding between presenting problem (and hence subsequent requirement for self-monitoring) and the person conducting the appointment. In the UK, practice nurses are often responsible for conducting chronic disease reviews, which may well have been the primary reason for consultation in those with diabetes. Current guidance regarding blood pressure management specifically in diabetes does not mention self-monitoring of BP and instead advocates repeat measurement at subsequent clinics when BP is high. We have, however, added some text to the discussion regarding this point:

“GPs were more likely to recommend home monitoring than nurses in those with diabetes. It is difficult to interpret this finding as it may reflect the primary reason for consultation, with certain tasks (such as diabetes reviews) performed primarily by nurses. Current guidance for BP management in diabetes recommends that high BP is confirmed at subsequent appointments, rather than through home monitoring.(33) Hence this finding, which importantly was not replicated in those without diabetes, may be explained if many of those with diabetes had annual review appointments. Overall, few patients were encouraged to monitor their BP at home, although it is likely that around 31% of patients were already self-monitoring based on previous UK survey data.(34)”

4. Figure 2 shows that with 2 measurements (blue line) the second reading is much lower than the 1st which would indicate that a third measurement is needed. The green line shows that the 1st and 2nd reading are almost similar and thus there would not be the need for a third measurement. This is probably related to the low number of patients but still this seems in contradiction to what one would expect (following the guidelines). What is the reason for this?

The reviewer is right that it is difficult to interpret the trends for second and third readings due to small numbers. We addressed the contradiction between the pattern observed in Figure 2 and our overall findings in our original discussion (“Although we have demonstrated that BP is measured in line with guidance in the majority of cases, this was driven by a large number of participants with low BP who had their BP measured only once”). We believe that if the patterns observed in Figure 2 can be confirmed in a larger study, they are most likely to be explained by a reluctance of clinicians to believe initially high readings, or those that require subsequent clinical action on the basis of incentive schemes. We have expanded some text in the discussion to make this point clear:

“Our results support the idea that professionals treat single readings with caution, particularly those above the diagnostic threshold, which require further action (e.g. in the form of treatment change). Although numbers were smaller, results suggest that this caution also extends to high BP sustained over two readings. Previous research suggests that recording of blood pressure may be influenced by specific BP related targets in the UK’s Quality and Outcomes Framework,(12) and hence routine practice in other healthcare systems, with different incentive schemes, may differ. Despite, agreement between current practice and guidelines, GPs may be better advised to use multiple readings more widely,(28) to ensure comparability with BP monitoring studies and detection of masked hypertension which affects approximately 19% of adults.(29).”

5. Discussion: The authors mention “UK” guidelines. However, I also know the guideline from the BHS. Do GPs generally follow NICE or also BHS guidelines? If my understanding is right, there are some differences between these guidelines.

The last guidance produced by BHS (now BIHS) alone was published in 2004 and recommended taking the average of at least two measurements which, as the reviewer states, differs from current NICE guidance. However, BIHS formally adopted the NICE guidance issued in 2006 and 2011, and the guidelines page on the BHS website (<https://bihsoc.org/guidelines/hypertension-management/>) provides links only to the current NICE guidance which was developed in collaboration with the BIHS. Hence any guidance previously issued by BIHS is superseded by NICE. Since the current NICE guidance was formally updated in 2011, with new evidence reviewed in 2016, we chose to make assessments based on the most up-to-date UK-based guidance.

6. The authors mention that the internet survey has the risk of underrepresenting certain groups such as e.g. the elderly (because of Internet). The survey requires a lot of reading and time to fill out which by itself also will lead to selection.

We agree with the reviewer that the population of patients that we recruited to collect data is selective. However, as we have argued in response to reviewer 1, since the purpose of this study was to examine the behaviour of family doctors and nurses, any threat to the validity of our results would arise due to selection bias in the population of doctors. It is not clear how any such biases at the patient-level would translate into biases at the healthcare professional level We have added further text to the discussion addressing this issue:

“Recruitment through patient involvement groups may have also resulted in over-representation of patients who are actively engaged with their healthcare, and due to phenomenon such as the inverse-care law,(24) may receive better quality (guideline adherent) care. However, since our aim was to study the behaviours of healthcare professionals, it is unclear how any biases at the patient-level will have translated into biases at the healthcare professional level. Furthermore, previous research regarding current practice of BP self-monitoring, showed similar results using both convenience and representative sampling of professionals.(25).”

7. As mentioned; considering guidelines for blood pressure measurement does not only entail the number of blood pressure measurements but also topics such as: was the right cuff size use, back support, no talking validated device measurement at heart level, no talking etc. etc. it is understandable that this could not all be asked but it is a limitation

We agree with the reviewer and have added text to the discussion to reflect this:

“Guidance covers many factors affecting the accuracy of BP measurement, such as the use an appropriately sized cuff, but such factors are less easily assessed by patients and we chose to limit the focus of this study in order to maximise response rates. The type and accuracy of devices used in

UK general practice has been studied previously,(27) but further direct observation of clinicians is warranted to determine if other aspects of BP measurement guidance is followed.”

8. Masked hypertension is shortly mentioned but might be good to give an estimated prevalence of masked hypertension based on previous studies. It might not be the aim of the paper but 50% of the patients without diabetes are measured only one time. I would think this is quite concerning. I think the authors agree with this : “ GPs may be better advised to use multiple readings more widely”. There is a reason that “Hypertension trials and epidemiological studies use multiple clinic blood pressure measurements at each visit”

Thank you for this suggestion. We have now added this information to the discussion and edited the text in question:

“Despite agreement between current practice and guidelines, GPs may be better advised to use multiple readings more widely,(28) to ensure comparability with BP monitoring studies and detection of masked hypertension which affects approximately 19% of adults.(29)”

VERSION 2 – REVIEW

REVIEWER	George Stergiou University of Athens School of Medicine Hypertension Center STRIDE-7 Third Department of Medicine Athens, Greece
REVIEW RETURNED	14-Feb-2018

GENERAL COMMENTS	No further comments
---------------------

REVIEWER	willem verberk Maastricht university Microlife AG Switzerland
REVIEW RETURNED	16-Feb-2018

GENERAL COMMENTS	I think the authors dealt very well with my comments and the comments of the second reviewer
--